# The Use of Botulinum Toxin A as an Adjunctive Therapy in the Management of Chronic Musculoskeletal Pain: A Systematic Review with Meta-Analysis

**DOI:** 10.3390/toxins13090640

**Published:** 2021-09-10

**Authors:** Simone Battista, Luca Buzzatti, Marialuisa Gandolfi, Cinzia Finocchi, Luca Falsiroli Maistrello, Antonello Viceconti, Benedetto Giardulli, Marco Testa

**Affiliations:** 1Department of Neurosciences, Rehabilitation, Ophthalmology, Genetics, Maternal and Child Health, University of Genova, 16132 Genova, Italy; simone.battista@edu.unige.it (S.B.); ft.luca.maistrello@gmail.com (L.F.M.); antonello.viceconti@gmail.com (A.V.); benedettogiardulli@gmail.com (B.G.); 2Department of Physiotherapy, Human Physiology and Anatomy, Experimental Anatomy Research Group, Vrije Universiteit Brussel (VUB), 1090 Brussels, Belgium; luca.buzzatti@vub.be; 3School of Allied Health, Anglia Ruskin University (ARU), Cambridge CB1 1PT, UK; 4Department of Neurosciences, Biomedicine and Movement Sciences, University of Verona, 37134 Verona, Italy; 5UOC Neurorehabilitation, AOUI Verona, 37134 Verona, Italy; 6Department of Neurosciences, IRCCS Ospedale Policlinico San Martino, 16132 Genova, Italy; cfinocchi@neurologia.unige.it; 7Department of Physical Medicine and Rehabilitation, AULSS9 Scaligera, G. Fracastoro Hospital, San Bonifacio, 37074 Verona, Italy

**Keywords:** botulinum toxins, type A, botulinum toxins, rehabilitation, physical and rehabilitation medicine, musculoskeletal pain, musculoskeletal disease, chronic pain, physical therapy modalities, physical therapy specialty, combined modality therapy

## Abstract

Several studies have investigated the effect of botulinum toxin A (BoNT-A) for managing chronic musculoskeletal pain, bringing contrasting results to the forefront. Thus far, however, there has been no synthesis of evidence on the effect of BoNT-A as an adjunctive treatment within a multimodal approach. Hence, Medline via PubMed, EMBASE, and the Cochrane Library-CENTRAL were searched until November 2020 for randomised controlled trials (RCTs) that investigated the use of BoNT-A as an adjunctive therapy for chronic musculoskeletal pain. The risk of bias (RoB) and the overall quality of the studies were assessed through RoB 2.0 and the GRADE approach, respectively. Meta-analysis was conducted to analyse the pooled results of the six included RCTs. Four were at a low RoB, while two were at a high RoB. The meta-analysis showed that BoNT-A as an adjunctive therapy did not significantly decrease pain compared to the sole use of traditional treatment (SDM −0.89; 95% CI −1.91; 0.12; *p* = 0.08). Caution should be used when interpreting such results, since the studies displayed very high heterogeneity (I = 94%, *p* < 0.001). The overall certainty of the evidence was very low. The data retrieved from this systematic review do not support the use of BoNT-A as an adjunctive therapy in treating chronic musculoskeletal pain.

## 1. Introduction

Musculoskeletal (MSK) pain is defined as acute or chronic pain that affects bones, muscles, ligaments, tendons, and nerves, and which arises from rheumatic MSK diseases (RMDs, e.g., osteoarthritis, low back pain, fibromyalgia) [1,2]. It is one of the leading causes of disability worldwide, with burdensome consequences on the health-related quality of life (HRQoL) of people with these diseases, as well as on the economic impact on society brought about by the healthcare system and insurance costs necessary for its management [1,3]. The prevalence of MSK pain in the general population is around 30%, ranging from 13.5% to 47% [3]. Several risk factors have been investigated for the development of MSK pain. These include age and gender, since older female adults have proved to be more prone to develop MSK pain [4], smoking, low levels of education, low levels of physical activity, poor social interaction, emotional distress, psychiatric disorders, a high workload, and poor familial assistance [3,5].

Patients with MSK pain present with a broad array of symptoms, including body aches, malaise, stiffness, fatigue, and sleep disorders [6,7,8,9]. The treatment of this condition, especially when chronic, should be based on a multimodal rehabilitation approach tailored to the patients’ needs and expectations [6,10,11,12]. This approach entails non-surgical treatments such as therapeutic exercise and patient education, as first-line intervention treatments, combined with acceptance and commitment therapy, manual therapy, and drug therapy (antidepressants and other drugs acting on central sensitisation processes), as conditional treatments, when appropriate [6,10,11,12]. However, the management of MSK pain conditions is complex and often presents challenges in the classification and diagnosis of these conditions, an overuse of surgery, and a low percentage of patients getting the proper care and education they should receive [6,13,14,15].

In the last few decades, several studies have investigated the use of botulinum toxin A (BoNT-A) in the management of chronic MSK pain in several RMDs (lateral epicondylitis, low back pain, plantar fasciitis, piriformis syndrome, etc.) [16]. BoNT-A is one of the several neurotoxins produced by *clostridium botulinum,* and it is usually used in the multidisciplinary management of spasticity in post-stroke cases, focal dystonia, and hyperkinetic disorders, for its ability to inhibit the release of the neurotransmitter acetylcholine at the neuromuscular junction, thus reducing striated muscle activity [17,18,19,20,21]. However, recent evidence has shown that BoNT-A also has an antinociceptive activity, indicating that it is characterised by a more complex mechanism of action than initially hypothesised, and which seems to add to its anticholinergic effect [22,23,24,25,26,27].

Studies have confirmed that BoNT-A can inhibit different nociceptive neurotransmitters, such as substance P (SP), calcitonin gene-related protein (CGRP), and excitatory neurotransmitters such as glutamate from sensory nerves [22,23,26,27]. Moreover, BoNT-A is also able to inhibit the translocation of diverse neurotransmitter receptors that are related to nociception, such as the N-methyl-D-aspartate (NMDA) receptor and transient receptor potential vanilloid 11 (TRPV1) [28,29]. Finally, Fos, a protein product of the gene c-fos, expressed after exposure to painful neuronal stimuli, was prevented by peripheral exposure to BoNT-A injection [30]. Altogether, this evidence suggests that BoNT-A can act on blocking peripheral sensitisation, which, in turn, may reduce central sensitisation, thus improving descended pain inhibitory control and cortical hyperexcitability [16,17,31,32,33,34].

These positive effects on pain mechanisms allowed BoNT-A to be used as a possible analgesic in people with chronic migraine, with and without medication overuse, other headache types, and chronic neuropathic pain [26,27,30,33]. The hypothesis that clinical conditions involving both peripheral and central sensitisation may benefit from BoNT-A treatment paved the way for its use in treating chronic MSK pain, which is characterised by a mixture of these two mechanisms [16,35]. In line with this, different studies have been carried out to understand if BoNT-A plays a role in managing chronic MSK pain. As a matter of fact, different and recent reviews suggest that BoNT-A can be effective in different RMD conditions such as chronic shoulder pain and osteoarthritis (OA) [23,24] but is ineffective in others, such as myofascial pain [25].

These studies, however, focus on the effect of BoNT-A in isolation and not as an adjunctive therapy for chronic MSK pain. Chronic MSK pain management is characterised by multimodal intervention that must include first-line interventions such as therapeutic exercise, patient education, and weight loss (when needed) [10]. Therefore, it is fundamental to investigate whether BoNT-A should be used as an adjunctive therapy to the routine management of chronic MSK pain rather than in isolation. The present systematic review is the first to investigate the use of BoNT-A as an adjunctive therapy in the management of chronic MSK pain in people with RMDs.

## 2. Results

### 2.1. Study Selection

The initial research yielded 2216 articles, which were reduced to 1795 after removing duplicates. After the screening selection through titles, abstracts, and full texts, 38 articles were screened for eligibility. Finally, six articles met the inclusion criteria and were included in this systematic review [36,37,38,39,40,41] (see Figure 1; PRISMA flow diagram). Only two studies openly disclosed having no conflict of interest concerning funding [36,37].

### 2.2. Study Characteristics

Table 1 reports the main characteristics of each study. The pooled population comprised 363 participants. Demographic information (e.g., sex and age) was retrieved only from five articles out of six, since one of them did not report this information [39]. In general, the studies were published between 2002 and 2019. One study conducted in China and in the USA focussed on knee osteoarthritis (N = 40) [36], two studies conducted in the USA investigated piriformis syndrome (N = 38) [39] and myofascial pelvic pain (N = 59) [37], one study conducted in Germany was directed to participants with cervical headache (N = 33) [41], and two studies carried out in Spain examined people with fibromyalgia (N = 44) [40] and cervicothoracic myofascial pain (N = 66) [38]. All the studies investigated the effectiveness in pain reduction of non-surgical interventions combined with BoNT-A against non-surgical interventions without BoNT-A. One study had three intervention groups with different BoNT-A doses, so that the pooled results among the different intervention groups were reported both for the study characteristics and outcomes [38].

As far as pain is concerned, four studies investigated it through a 0–10 VAS scale [36,37,39,40], one study with a 0–100 point scale [41], and one with a 0–70 VAS scale [38]. The HRQoL (secondary outcome) was evaluated in three studies out of six [36,38,40], with different scales. Two studies used the Medical Outcomes Study 36-item Health Survey (SF-36) [36,38], and one study used the EuroQol-5D Health Questionnaire (EQ-5D) [40].

### 2.3. Risk of Bias in Studies

The risk of bias assessment is displayed in Figure 2. Among the included studies, four were at a low risk of bias, while two were at a high risk. One study was assessed at high risk because of missing outcome data, since the data was not available for all the participants included in the study, and missing outcome data was not adjusted through analysis methods that correct for bias or sensitivity analysis [39]. Moreover, no reason was given for the missing values [39]. The other study was assessed as being at high risk because the baseline differences between groups highlighted possible issues in the randomisation process adopted since the percentage of older individuals was significantly different in the different groups of the study [40].

### 2.4. Results of Individual Studies

The results of individual studies are presented in Table 2 and hereafter discussed. When available, the results are reported at baseline, and after two, four, eight, and twelve weeks. As far as any adverse effects regarding the use of BoNT-A are concerned, two studies declared that patients did not report any side effects [36,41], and one stated that four participants reported de novo constipation and urinary incontinence, and other complications including recurrent urinary tract infection, faecal incontinence, and urinary retention, each occurring in fewer than four participants [37]. However, the frequency of these side effects was not statistically different compared to the placebo group that received a saline injection. One study did not mention any side effect [38]. One study reported that two participants experienced injection site pain, one participant had flu-like symptoms, one participant reported having a stiff neck, and one participant reported having a wobbly neck [39]. One study declared that the participants in the intervention group experienced more side effects compared to the control group, but these were not reported [40].

#### 2.4.1. Primary Outcome—Pain

As far as pain reduction is concerned, two studies found that the intervention group significantly reduced this outcome compared to the control group [36,39]. In the other four studies, no significant differences were found between the control and intervention groups [37,38,40,41].

#### 2.4.2. Secondary Outcome—HRQoL

In the three studies that evaluated HRQoL, one study found a significant difference between the control and intervention groups, both regarding the physical and mental components of the SF-36 questionnaire [36]. One study reported that the intervention group demonstrated an improvement in the role-emotional subscale of the SF-36 as compared with the controls (*p* < 0.05), while a trend towards improvement was seen in the vitality (*p* = 0.053) and social functioning (*p* = 0.057) subscales [38]. Unfortunately, this study did not report raw results. Finally, one study did not find any significant difference in HRQoL between the two groups mentioned earlier [40].

### 2.5. Results of Syntheses

A total of five articles had data available for meta-analysis, whereas the authors of the sixth study were contacted and provided the missing data needed [37]. Figure 3 summarises the results of the meta-analysis. In order to guarantee the same follow-up for all studies, the meta-analysis was performed using the results at week eight from the baseline. The pooled results show that BoNT-A, in addition to multimodal management of RMDs, did not significantly decrease pain compared to not adding it in the multimodal treatments used as usual care (SDM −0.89; 95% CI −1.91; 0.12; *p* = 0.08). Studies displayed considerable heterogeneity (I = 94%, X^2^ test < 0.001).

Subgroup analysis was not conducted because of the different diseases treated in the various studies, the different BoNT-A dosages, and the different RMD management strategies adopted.

### 2.6. Reporting Biases

Two studies had no raw data regarding the general population characteristics [42] and the HRQoL outcome investigated [38]. It was not possible to assess the publication bias in detail since a funnel plot could not be generated due to the low number of studies (<10), as reported in the Cochrane Handbook for Systematic Reviews of Interventions [43]. Finally, unpublished studies and studies not written in English were not considered.

### 2.7. Certainty of Evidence

The overall certainty of the evidence was very low, as assessed through the GRADE approach. In Table 3, the whole GRADE assessment is reported.

## 3. Discussions

In the last few decades, BoNT-A has been used to treat several conditions, including chronic MSK pain, thanks to its antinociceptive effects. Some authors have investigated its positive effect combined with different rehabilitative procedures (e.g., manual therapy, extracorporeal shock wave therapy, electrical stimulation) in post-stroke spasticity and chronic migraine [18,44]. However, only a few studies have investigated the value of BoNT-A as an adjunctive therapy in managing chronic MSK pain in people with RMDs. From what was retrieved in this study, adding BoNT-A does not seem to bear significant effects in managing chronic MSK pain.

Only two studies reported a significant difference in adding BoNT-A in the management of chronic MSK pain [36,39]. In these two studies, BoNT-A injection was used as an adjunctive therapy in a protocol based on active interventions (i.e., therapeutic exercise). Once injected, BoNT-A has a “sphere” of diffusion because it spreads in all directions from an injection site. Not only is this “sphere” determined by the pharmacological parameter (i.e., dosage), but also by the treatments with which the BoNT-A injection is combined. A recent review has shown that adjuvant treatments (e.g., electric stimulation) may enhance the internalisation of this drug [18], though little is known about the role of therapeutic exercise in the process of the internalisation of BoNT-A. Nonetheless, previous evidence highlighted that exercise, in general, has no substantial effect on the absorption of orally administered drugs, yet it appears to enhance absorption from intramuscular, subcutaneous, and transdermal sites [45,46]. In line with this, it is possible to hypothesise that exercise plays a role in the internalisation of BoNT-A. Further studies should investigate this association, since exercise represents one of the first-line interventions in the management of chronic MSK pain [10,11].

In contrast to what was mentioned above, one study which focussed on myofascial pelvic pain did not find a significant effect of BoNT-A injection combined with exercise [37]. However, in this study, BoNT-A injection and therapeutic exercise did not start simultaneously but four weeks after the first injection. Moreover, in this study, the participants had multiple pain disorders, and the study population was based on women with severe refractory pain [37]. Future studies should test the combination of exercise therapy and BoNT-A when started simultaneously in a cohort of participants with localised pain and with several degrees of symptomatology.

As far as myofascial pain syndromes are concerned, a recent Cochrane Systematic Review investigated the effect of BoNT-A injection in this type of chronic MSK pain [25]. In this review, the authors concluded that there was inconclusive evidence to support the use of BoNT-A injection in this type of syndrome. Currently, even if there is a scientific rationale behind the use of BoNT-A to treat trigger points (i.e., blocking the acetylcholine path), there are contradictory results regarding the efficacy of this substance in the treatments of myofascial pain syndromes [47,48]. Through the present review, it is possible to reinforce this evidence, confirming the fact that BoNT-A seems not to provide further improvement in a multimodal approach. The studies that dealt with myofascial pain syndromes did not find any differences in the pain outcome when adding BoNT-A compared to a placebo [37,38]. However, these two studies used a saline injection in the control group, and there is evidence that support the use of saline injection in myofascial pain, which contributes to considering saline injection as a real treatment for myofascial pain syndromes because of the injection of dry needles [49]. As a result, dry needling (i.e., the use of dry needles to deactivate trigger points) has already been shown to decrease pain in patients with these syndromes [50].

Two studies investigated the use of BoNT-A injection in addition to patient education and physical therapy modalities in fibromyalgia and cervicogenic headache, respectively. Furthermore, these two studies found that there were no significant differences between the intervention group and the control group. This is in line with other studies investigating the use of BoNT-A in the abovementioned diseases [19,51]. However, as for other RMDs, fibromyalgia and cervicogenic headache need a multimodal approach that cannot be limited to drug and passive therapies, but that is centred on lifestyle changes and exercise [10]. Future studies should consider this multimodal approach based on active treatment to investigate whether adding BoNT-A may have a role in treating these diseases.

Some methodological issues among the included studies may explain the inconsistency of the reported effect of BoNT-A as an adjunct treatment in the management of chronic MSK pain. Firstly, although the origin of pain among the different studies was similar (chronic MSK pain), it is to be noted that it stemmed from different diseases. Secondly, the studies adopted different types of injections (intramuscular, intra-articular etc.) and different dosages. Thirdly, the chronic MSK pain management of the studies was heterogeneous, with some studies adopting active strategies and others focussing more on passive interventions. Besides the robust methodology used to conduct it, the strengths of this review are the fact all the included studies shared a common and evidence-based definition of chronic MSK pain to define the enrolled population and that they studied the effect of multidisciplinary management of chronic MSK pain, which is the gold-standard of chronic pain management, and not the effect of a treatment in isolation. Finally, all the included studies reported very few adverse effects, which confirms the safety of using BoNT-A when treating these diseases. 

Taken altogether, the data retrieved from this systematic review, as synthesised through the meta-analysis, do not support BoNT-A as an adjunctive therapy in the treatments of chronic MSK pain. The reasons behind the failure of BoNT-A might stem from different causes, both clinical and methodological. The former depend on the fact that chronic MSK pain is a complex and multifaceted symptom whose source and dysfunction can be difficult to determine. Its pathogenic mechanisms can be related to nociceptive (e.g., sensory innervation of bones, joints, and muscles), neuropathic (e.g., the interplay of neurons and non-neural cells), and nociplastic (i.e., cortical remodelling induced by chronic pain) phenomena [52,53]. As mentioned above, BoNT-A seems to play a role in pain modulation via nociceptive and neuropathic mechanisms, reducing the peripheral sensitisation, thus also having a possible effect on the central one [16,17,31,32,33,34]. However, this effect may not be enough to induce a real modification at the cortical level, which is one of the main aims of the treatment of this condition. Moreover, each pathological condition may be characterised by one or more of the abovementioned pathogenic mechanisms. Nevertheless, none of the included studies in this systematic review investigated the pathogenic mechanism underlying patients’ pain perception, therefore identifying adequate pain subgroups that might have benefitted from BoNT-A injections rather than others.

This issue allows the discussion to move towards the latter reasons behind the unsuccessful effect of BoNT-A in the management of chronic MSK pain, that is to say, the methodological problems behind the design of the included studies. The first one has just been discussed, and it relates to the missed investigation of patients’ prevalent pain mechanisms. The second one relates to the BoNT-A dosage adopted. As reported by Lim et al., the benefits derived from BoNT-A injections may be negated by incorrect injection of BoNT-A in the targeted site [54]. The authors stated that several elements influenced BoNT-A internalisation, such as the dose, concentration, number, site and rate of injections, needle gauge, muscle size, the distance of the needle tip from the neuromuscular junction, and the protein content of the formulation [54]. The use of BoNT-A in chronic MSK pain is new, and no clinical practice guidelines have been developed so far. This may lead to incorrect use of BoNT-A that is left to the clinicians’ preferences and beliefs and that is not based on an evidence-based approach.

Finally, as it can be seen from the GRADE approach, the certainty of this evidence is very low. The scientific literature needs more studies, with a more accurate design, so as to draw further conclusions on the use of BoNT-A in the management of chronic MSK pain. Besides, although active therapies are now the core treatment of RMDs, only a few studies deployed them in their design. Regardless of the use of BoNT-A, patients with RMDs need to be instructed on the fact that passive modalities alone will fail to change their symptomatology, above all in the chronic phase in which maladaptive learning mechanisms to deal with pain have risen, such as kinesiophobia and fear avoidance [5]. Other passive treatments, including BoNT-A, need to be reconceptualised to support first-line interventions and not as core treatments themselves, integrated into a multimodal approach that will enhance patients’ symptomatology and HRQoL.

## 4. Materials and Methods

This systematic review protocol was registered into the International Prospective Register of Systematic Reviews (PROSPERO; No. CRD42021231140). The Cochrane Handbook for Systematic Reviews of Interventions was used to develop, implement, and conduct this systematic review [43]. The reporting of this systematic review followed the Preferred Reporting Items for Systematic Reviews and Meta-Analyses statement (PRISMA) 2020 [55].

### 4.1. Eligibility Criteria

#### 4.1.1. Types of Studies

Among the different study designs, only randomised controlled trials (RCTs) were taken into account. Thus, non-RCTs, case series, and single-case studies were excluded from the analysis. Moreover, only published journal articles written in English were considered eligible. Finally, limitations on the publication date were set at studies published after 2000. Hence, all the RCTs published from January 2020 to 1 December 2020 were considered eligible.

#### 4.1.2. Types of Participants

Studies addressing only adults (age > 18 years) with chronic MSK pain were considered eligible for this systematic review. In order to consider the MSK pain as chronic, this study adopted the definition provided by the ICD 11: “persistent or recurrent pain that arises as part of a disease process directly affecting bone(s), joint(s), muscle(s), or related soft tissue(s)” [56]. No restrictions were imposed on the sex as well as the gender of the participants. Furthermore, no follow-up and symptom duration limits were imposed. Conversely, studies that included individuals with central or peripheral neurologic acute and chronic diseases (e.g., peripheral neuropathy, stroke, dystonia, infantile palsy, spasticity), acute pain, or specific conditions such as cancer, hyperhidrosis, otitis, bruxism, and pelvic pain induced by urogynecological diseases were excluded.

#### 4.1.3. Types of Interventions

Studies that investigated BoNT-A as an adjunctive therapy in the management of chronic MSK pain were considered eligible. Thus, in order to be included, studies needed to compare the effectiveness of other non-surgical interventions for chronic MSK pain with and without the combined use of BoNT-A. Non-surgical interventions for the management of chronic MSK pain were as follows: therapeutic exercises (e.g., stretching or reinforcement), manual techniques directed to muscles (e.g., myofascial tissue techniques), patient education, articular joint techniques (e.g., mobilisation or manipulation techniques), peripheral nerve mobilisations (e.g., neurodynamics), drugs, physical therapies, and psychological interventions. Studies that evaluated BoNT-A in isolation were, therefore, excluded.

#### 4.1.4. Types of Outcome Measures

The primary outcome of this study was self-reported pain perception (e.g., VAS and NRS), and the secondary outcome of this study was health-related quality of life (HRQoL).

### 4.2. Information Sources

Electronic databases including Medline via PubMed, EMBASE, and the Cochrane Library-CENTRAL were systematically searched up to 30 November 2020.

### 4.3. Search Strategy

The search was conducted by two authors (S.B. and L.B.). The search strategy was a combination of Medical Subjects Headings and the keywords “botulinum toxin, type A”, “musculoskeletal pain”, “myofascial pain syndromes”, “low back”, “dysfunctions”, “rheumatic diseases”, “trigger points”, “tendinopathy”, “fibromyalgia”, “osteoarthritis”, “neck pain”, “cervicogenic headache”, and “randomised controlled trials”, and a combination thereof. Free researches through scientific websites (i.e. sciencedirect.com) and the references lists of retrieved articles screened by full text were also performed.

### 4.4. Selection Process

Articles retrieved were uploaded onto the Rayyan website [57,58] after duplicate removal. Afterwards, two researchers (A.V. and B.G.) independently and systematically carried out the initial search applying the inclusion and exclusion criteria to titles and abstracts. When necessary, the full text was read. A third author (S.B.) was consulted in case of disagreement between reviewers to reach a consensus. No authors or experts were contacted to get additional studies.

### 4.5. Data Collection Process

Two researchers (C.F. and L.F.M.) independently extracted the following data from each study using standardised Excel templates: first author, year of publication, country, setting, study design, disease of interest, type of intervention and control, the total number of participants, age, number of females/males, number of groups, number in each group, the timing of administration of intervention and baseline, post-intervention, and follow-up (when available) point estimates and measures of variability of main outcomes. Results for both primary and secondary outcomes were extracted. The corresponding authors of studies where data were not completely displayed were contacted. Disagreements in the data collection process were resolved by either a consensus process or consultation with a third author (S.B.).

### 4.6. Study Risk of Bias Assessment

The risk of bias and methodological quality of the included randomised controlled trials were independently assessed by two researchers (C.F. and L.F.M.) using the Revised Cochrane risk-of-bias tool 2.0 (RoB 2.0), which is the recommended tool to assess the risk of bias in randomised trials included in Cochrane Reviews [59]. This tool allows for assessing the risk of bias based on a standard set of items used for the risk of bias appraisal: “bias arising from the randomisation process”, “bias due to deviations from intended interventions”, “bias due to missing outcome data”, “bias in the measurement of the outcome”, “bias in the selection of the reported result”, and, finally, the risk of bias judgment for each outcome. Through RoB 2.0, studies were classified as “low”, “some concerns”, or “high” risk of bias. In case of disagreement between the two abovementioned reviewers, a consensus was obtained after consulting a third author.

### 4.7. Data Analysis and Synthesis

Statistical analysis was calculated by Review Manager 5.3 (RevMan–Copenhagen: The Nordic Cochrane Center, The Cochrane Collaboration, 2014) and Stata 16 (StataCorp). For inter-group comparisons, the mean, standard deviation, and/or mean differences for pre- and post-treatment conditions and the 95% CI to compare intervention and control group effects were extracted or calculated. In the case of multiple intervention groups with a single control group, pooled results of the intervention groups were computed as recommended in the Cochrane Handbook [43].

Continuous data were combined through meta-analysis and analysed using the inverse variance statistical method, with a random effect model. Due to different measures to assess the same primary outcome, the standardised mean difference was chosen as the effect measure. Statistical heterogeneity was assessed using the chi^2^ test and the I^2^ statistic. An I^2^ from 0% to 40% indicated not important heterogeneity; 30% to 60% moderate heterogeneity; 50% to 90% substantial heterogeneity; and 75% to 100% considerable heterogeneity. In this systematic review, studies displayed considerable heterogeneity (I = 94%, X^2^ test < 0.001), and thus caution should be used to interpret such results.

The overall quality of the evidence and strength of the recommendations were evaluated using the GRADE approach [60] through the GRADEpro GDT (https://gradepro.org/, accessed date: 15 July 2021), which was applied for the primary outcome included in the meta-analysis (pain). The downgrading process was based on five domains: study limitations (e.g., risk of bias), inconsistency (e.g., heterogeneity between study results), indirectness of evidence (generalisability and transferability, e.g., short-term follow-up), imprecision (e.g., small sample size), and reporting bias (e.g., publication bias).

## Figures and Tables

**Figure 1 toxins-13-00640-f001:**
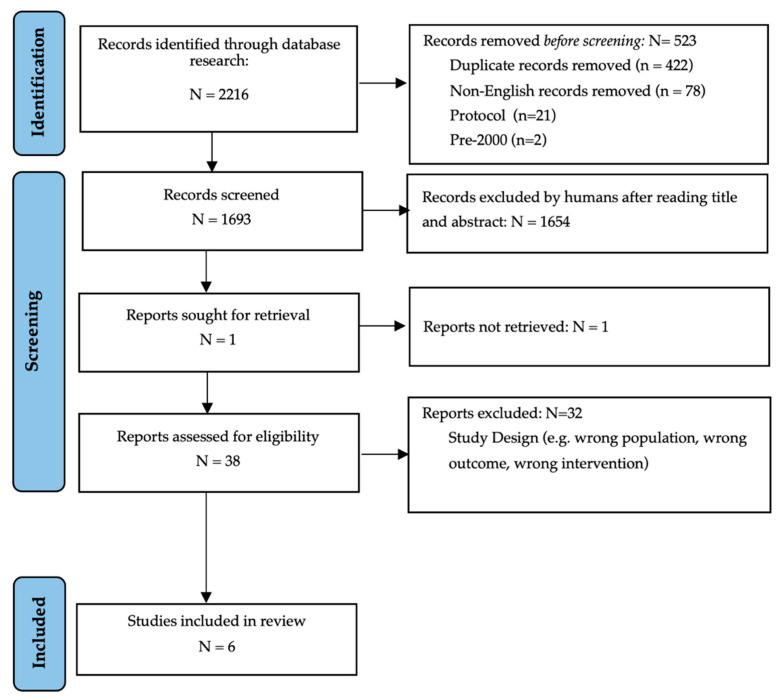
PRISMA 2020 flow diagram.

**Figure 2 toxins-13-00640-f002:**
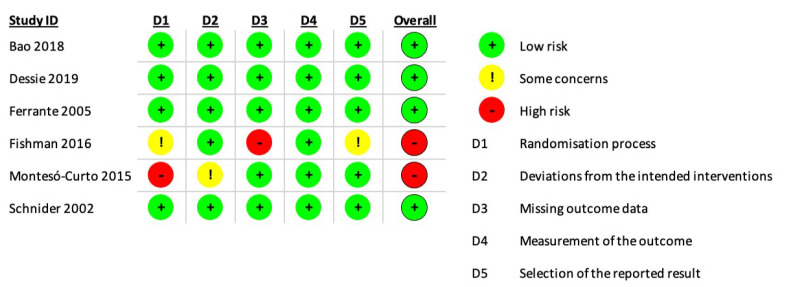
Risk of bias assessment of the studies included.

**Figure 3 toxins-13-00640-f003:**
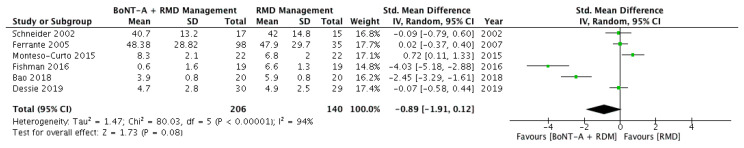
Meta-analysis for the primary outcome (pain) at week 8.

**Table 1 toxins-13-00640-t001:** Characteristics of the included studies.

Author, Year	Setting	Disease	Demographic Characteristics	Intervention	Control	Physiotherapy Intensity
Dessie 2019	Outpatient office	Myofascial pelvic pain	N = 59 Groups and sex: 30 Int. (30 F, 0 M) 29 Cont. (29 F, 0 M) Age: (Median (Q1,Q3)) Int. 42 (30,55) Cont. 40 (31,54) Dropout: No dropouts	20 BoNT-A injections (200 U) + Manual therapy (individualised abdominal soft and connective tissue targeted manual therapy) + Patient education + Exercise regarding movement, biopsychosocial pain concepts, alignment, and respiration	20 saline injections (1 mL) + Manual therapy (individualised abdominal soft and connective tissue targeted manual therapy) + Patient education + Exercise regarding movement, biopsychosocial pain concepts, alignment, and respiration	1 s/w for 8 weeks. Each session lasted 45–60 min.
Bao 2018	Rehabilitation department	Knee osteoarthritis	N = 40 Groups and sex: 20 Int. (10 F, 10 M) 20 Cont. (11 F, 9 M) Age: (Mean ± SD) Int. 66 ± 3.5 Cont. 65 ± 3.52 Dropout: No dropouts	BoNT-A injection (100 U) + Therapeutic exercise (strengthening and balance)	Saline injection (2.5 mL) + Therapeutic exercise (strengthening and balance)	5 s/w for 8 weeks. Each session lasted 30–45 min.
Fishman 2016	Private clinic	Piriformis syndrome	N = 54 Groups and sex: 27 Int. (NA F, NA M) 27 Cont. (NA F, NA M) Age NA Dropouts: 52 week 2 (25 Int., 27 Cont.) 49 week 4 (25 Int., 24 Cont.) 38 week 8 (19 Int., 19 Cont.) 15 week 12 (9 Int., 6 Cont.)	4 BoNT-A injections (300 U) + Physical therapies (US, hot pack, cold spray) + Manual therapy (myofascial release, stretching) + Therapeutic exercises (McKenzie exercises)	4 saline injections (3 mL) + Physical therapies (US, hot pack, cold spray) + Manual therapy (myofascial release, stretching) + Therapeutic exercises (McKenzie exercises)	1 s/w for 12 weeks
Montesó-Curto 2015	Hospital	Fibromyalgia	N = 44 Groups and sex: 22 Int. (21 F, 1 M) 22 Cont. (21 F, 1 M) Age: N (%) N < 60 y, 22 (50%) N ≥ 60 y, 22 (50%) Dropout: No dropouts	BoNT-A injections (unit not available) + Patient education	Patient education	4 sessions in total: 1 s/w per 3 weeks 1 session after 1 month from the third one
Ferrante 2005	Unknown	Cervicothoracic myofascial pain	N = 133 Groups and sex: 98 Int. (61 F, 37 M) 35 Cont. (20 F, 15 M) Age: (Mean ± SD) Int. 45.5 ± 14.8 Cont. 45.3 ± 10.1 Dropout: No dropouts	BoNT-A injection (10/25/50 U per TrP up to maximum of 5 TrPs) + Manual therapy (myofascial release) + Pharmacotherapeutic regimen (amitriptyline, ibuprofen, propoxyphene–acetaminophen napsylate)	Saline injections (unit not available) + Manual therapy (myofascial release) + Pharmacotherapeutic regimen (amitriptyline, ibuprofen, propoxyphene–acetaminophen napsylate)	No data available
Schnider 2002	Unknown	Cervical headache	N = 33 Groups and sex: 17 Int. (10 F, 7 M) 16 Cont. (10 F, 6 M) Age: (Mean ± SD) Tot. 50.7 ± 10.4 Dropout: No dropouts	BoNT-A injections (15 U each for 6 most painful TrPs) + Manual therapy (massage) + Physical therapy (hot packs)	Saline injections (0.15 mL each for 6 most painful TrPs) + Manual therapy (massage) + Physical therapy (hot packs)	9 sessions in total divided into 3 weeks

Legend: N, number; Int., intervention; Cont., control; F, female; M, male; Q1; first quartile; Q3, third quartile; SD, standard deviation; NA, not available; y, years old; Tot, total group; BoNT-A, botulinum toxin A; U, unit; s/w, session per week; US, ultrasound; TrP, trigger point.

**Table 2 toxins-13-00640-t002:** Primary (pain) and secondary outcomes (health-related quality of life) of the included studies.

Author, Year	Outcome Measure	Group	Baseline	T1	T1 CI 95%	T2	T2 CI 95%	T3	T3 CI 95%	T4	T4 CI 95%
**Pain (Primary Outcome)**											
Dessie 2019	VAS (0–10)	Int.	5.2 ± 2.6	4.7 ± 2.8 ^‡^	0.0 [−1.6;1.3]						
		Cont.	5.2 ± 2.4	4.9 ± 2.5 ^‡^							
Bao 2018	VAS (0–10)	Int.	6.8 ± 1.1	4.7 ± 0.8 ^†^	−1.2 [−1.7;−0.7]	3.9 ± 0.8 ^‡^	−2.0 [−2.5;−1.5]				
		Cont.	6.6 ± 0.8	5.9 ± 0.7 ^†^		5.9 ± 0.8 ^‡^					
Fishman 2016	VAS (0–10)	Int.	7.1 ± 1.8	3.5 ± 2.3 *	−2.7 [−3.9;−1.5]	3.2 ± 3.1 ^†^	−2.9 [−4.3;−1.5]	0.6 ± 1.6 ^‡^	−6.0 [−7.0;−5.0]	0.9 ± 2.4 ^§^	−5.6 [−7.4;−3.8]
		Cont.	6.6 ± 2.0	6.2 ± 1.9 *		6.1 ± 1.2 ^†^		6.6 ± 1.3 ^‡^		6.5 ± 0.2 ^§^	
Monteso-Curto 2015	VAS (0–10)	Int.	7.4 ± 2.1	8.3 ± 2.1 ^‡^	1.4 [−7.9;10.7]						
		Cont.	6.5 ± 1.9	6.8 ± 2.0 ^‡^							
Ferrante 2005	VAS (0–70)	Int.	63.2 ± 22.0	60.7 ± 24.3 *	10.5 [−0.5;21.6]	53.3 ± 26.3 ^†^	7.2 [−3.9;18.3]	48.4 ± 28.8 ^‡^	0.5 [−11.1;12.1]	51.1 ± 26.9 ^§^	1.8 [−10.7;14.3]
		Cont.	59.7 ± 24.4	50.2 ± 29.2 *		46.1 ± 28.9 ^†^		47.9 ± 29.7 ^‡^		49.3 ± 33.1 ^§^	
Schnider 2002	VAS (0–100)	Int.	52.6 ± 13.6	42.0 ± 14.8 ^‡^	1.3 [−8.6;11.2]						
		Cont	50.3 ± 13.2	40.7 ± 13.2 ^‡^							
**HRQoL (Secondary Outcome)**											
Bao 2018	PCS 36 (0–100)	Int.	37.3 ± 7.5	47.3 ± 10.1 ^†^	10.7 [4.9;16.5]	56.9 ± 11.2 ^‡^	20.4 [14.4;26.4]				
		Cont.	36.2 ± 9.9	36.6 ± 7.9 ^†^		36.5 ± 6.9 ^‡^					
	MCS 36 (0–100)	Int.	43.6 ± 7.7	54.3 ± 8.0 ^†^	9.3 [4.3;14.3]	61.7 ± 9.0 ^‡^	16.8 [11.6;22.0]				
		Cont.	44.5 ± 7.73	45.0 ± 7.7 ^†^		44.9 ± 7.2 ^‡^					
Monteso-Curto 2015	EQ-5D (0–5)	Int.	4.0 ± 1.9	4.2 ± 2.4 ^‡^	−0.1 [−2.2;0.4]						
		Cont.	4.3 ± 1.6	5.1 ± 1.8 ^‡^							
Ferrante 2005	SF-36 (0–100)	Int.	No data								
		Cont.	No data								

Legend: 95% CI, 95% confidence interval; VAS, visual analogue scale; HRQoL, health-related quality of life; PCS, physical component summary (physical functioning, role-physical, and bodily pain); MCS-36, mental component summary (social functioning, role-emotional, and mental health); EQ-5D, EuroQol-5D Health Questionnaire; SF-36, Short Form Health Survey 36 items; Int., intervention; Cont., control; * two weeks; ^†^ four weeks; ^‡^ eight weeks; ^§^ twelve weeks. The authors calculated all CIs at 95%.

**Table 3 toxins-13-00640-t003:** GRADE approach assessment.

Certainty Assessment							№ of Patients		Effect		Certainty	Importance
**№ of Studies**	**Study Design**	**Risk of Bias**	**Inconsistency**	**Indirectness**	**Imprecision**	**Other Considerations**	**BoNT-A + Non-Surgical Interventions**	**Non-Surgical Interventions**	**Relative** **(95% CI)**	**Absolute** **(95% CI)**		
**Pain** (**Follow-Up: Mean Eight Weeks; Assessed with: VAS; See in Table 2 the Ranges Adopted in the Studies for the VAS Scales**)												
6	RCT	serious ^a^	very serious ^b^	not serious	serious ^c^	none	139	140	/	SMD 0.89 SD lower (1.91 lower to 0.12 higher)	⊕◯◯◯† VERY LOW	CRITICAL

Legend: BoNT-A; botulinum toxin A; CI, confidence interval; SMD, standardised mean difference; ^a^ downgraded one level due to bias from missing values and randomisation process; ^b^ downgraded two levels due to a considerable heterogeneity of the studies and substantial inconsistency among them (BoNT-A dose, interventions, etc.); ^c^ downgraded one level due to low sample size and contradictory results; † The GRADE approach uses different ⊕ to declare the level of certainty: one ⊕ means very low level of certainty (as in this review), two ⊕ means low, three ⊕ stands for moderate, four ⊕ stands for high.

## Data Availability

The datasets used and analysed during the current study are available from the corresponding author on reasonable request.

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
