# Peer review of "The Use of Botulinum Toxin A as an Adjunctive Therapy in the Management of Chronic Musculoskeletal Pain: A Systematic Review with Meta-Analysis"

_toxins, 2021, doi:10.3390/toxins13090640_

Round 1
Reviewer 1 Report
This is an important topic. The newly discovered antinociceptive properties of Botulinum toxin A sheds lights on the novel use of the toxin in pain. However the previous systematic reviews and metanalyses indicated the fact that the toxin is useful in neuropathic pain. To my knowledge this systematic review with meta-analysis comprehensibly focuses on the of the usefulness of the toxin in chronic musculoskeletal pain.
Suggestion- You could have added a funnel plot assessing the publication bias for the final continuous VAS outcome.
Reviewer 2 Report
This is an interesting and unique angle of looking at adjunctive therapy of BoNT for MSK pain. It is very important topic to complement numerous systematic reviews published also showing negative results of BoNT-A for MSK pain.
There are however significant flaws to this manuscript and this manuscript requires extensive major revisions.
Firstly, the title is not clear and I am not sure why the title is using onabotulinum toxin A- rather than Botulinum Toxin A- as studies that were not Onabotulinum toxin were also assessed in this systematic review.
Also, the term "Adjunctive Therapy" should be used as this is the focus of this systematic review. The term in the title:" Added value of .." is not clear to the reader that the investigation is Adjunctive therapy of BoNT-A to conservative treatment for MSK pain.
Also a search of the term "Adjunctive therapy" is extremely important when conducting the literature search. This search term is a key term for this systematic review and without this term the literature search will be incomplete.
The Introduction is not clear and needs major revision. More has to be discussed regarding the concept of central pain sensitization and how BoNT-A could theoretically work on peripheral and central processing of pain. Also there is significant literature describing the neuropathic element of pain in OA and MSK pain. The neuropathic pain guidelines from Lancet neurology 2016 (Nadine Attal) should also be included as BoNT-A is included in treatment of neuropathic pain- and this could make discussion more interesting of why BoNT-A may be used.
There have been significant and high caliber systematic reviews assessing the role of BoNT-A in MSK pain and these have to be described including reviews by:
Cochrane review by A. Soares. doi.org/10.1002/14651858.CD007533
Systematic review in Toxins for BoNT-A for Shoulder MSK pain, and for chronic joint pain:
Po-Cheng Hsu et al. Toxins 2020. 12, 251; doi/10.3390/toxins 12040251
Nicole Blanshan, Hollis Krug. The Use of Botulinum Toxin for the Treatment of Chronic Joint Pain: Clinical and Experimental Evidence.
The reason that these need to be addressed is to show the reader that systematic reviews have been conducted on this subject but that the uniqueness of your study is that it is looking at adjunctive therapy of BoNT to MSK.
The results, discussion and limitations section needs to be clearer. It would be important to discuss the caliber of the studies either ranked by Pedro scale or Sackett's level of evidence. This will make it clear to the reader the level of the study selected.
The discussion needs to be more robust. It needs to discuss the complexity of MSK pain- as neuropathic pain and nociceptive element. It should also discuss why BoNT is not showing effectiveness. Also discuss whether there were any adverse effects post BoNT injections (weakness for example)- as this can weaken muscle and possibly contribute to functional decline and biomechanical changes.
Author Response
Please, see the attachment.

Round 2
Reviewer 2 Report
This manuscript is now more robust after the revisions.
I would recommend that the abstract be clearer- it is very wordy and can be simplified and to the point. The key points can also be clarified with less wordiness and be more succinct- this will give higher impact to your manuscript.
I am very happy with all the revisions but would recommend adding a sentence about the adverse event profile (BoNT-A+adjunctive therapy) stated in the RCTs that have been chosen for this systematic review. It is important to have this for the reader to understand that there were very few adverse effects reported.
I am happy with the response regarding not using Sackett/Pedro and it would be of interest to our readers to explain why you used the grading system in the manuscript- as you stated in your response to me.
I would recommend going through the manuscript for syntax and fine tuning the "wordiness"- I am not sure if it is a translation issue but some sentence structures are difficult to follow. For example: lines 315-318 is a long sentence structure and I understand what you are saying but not clear in its written form.
I have a suggestion- as you use the term musculoskeletal throughout the text whether you use the abbreviation (MSK) throughout the text-so text will then look less wordy.
Author Response
Please, see the attachment.
